# Kinetics of Human Serum Albumin Adsorption on Polycation Functionalized Silica

**DOI:** 10.3390/biom14050531

**Published:** 2024-04-29

**Authors:** Małgorzata Nattich-Rak, Dominik Kosior, Maria Morga, Zbigniew Adamczyk

**Affiliations:** Jerzy Haber Institute of Catalysis and Surface Chemistry, Polish Academy of Sciences, Niezapominajek 8, PL-30239 Krakow, Poland; malgorzata.nattich-rak@ikifp.edu.pl (M.N.-R.); maria.morga@ikifp.edu.pl (M.M.)

**Keywords:** adsorption of HSA, albumin adsorption, HSA layers at silica, human serum albumin adsorption, kinetics of HSA adsorption, poly-L-arginine, optical reflectometry, silica sensors

## Abstract

The adsorption kinetics of human serum albumin (HSA) on bare and poly-L-arginine (PARG)-modified silica substrates were investigated using reflectometry and atomic force microscopy (AFM). Measurements were carried out at various pHs, flow rates and albumin concentrations in the 10 and 150 mM NaCl solutions. The mass transfer rate constants and the maximum protein coverages were determined for the bare silica at pH 4.0 and theoretically interpreted in terms of the hybrid random sequential adsorption model. These results were used as reference data for the analysis of adsorption kinetics at larger pHs. It was shown that the adsorption on bare silica rapidly decreased with pH and became negligible at pH 7.4. The albumin adsorption on PARG-functionalized silica showed an opposite trend, i.e., it was negligible at pH 4 and attained maximum values at pH 7.4 and 150 mM NaCl, the conditions corresponding to the blood serum environment. These results were interpreted as the evidence of a significant role of electrostatic interactions in the albumin adsorption on the bare and PARG-modified silica. It was also argued that our results can serve as useful reference data enabling a proper interpretation of protein adsorption on substrates functionalized by polyelectrolytes.

## 1. Introduction

Human serum albumin (HSA) finds many practical application for the preparation of targeted drug delivery systems and for coating of medical devices: hemodialyzer membranes, endovascular stents, synthetic vascular grafts, pacemakers, orthopedic titanium implants and catheters that can prevent adhesion of other proteins, platelets and bacteria [1,2,3]. It is also commonly used as blocking agents by preparing immunoglobulin-covered polymer particles used in agglutination immunoassays [4,5]. The albumin is also used for the preparation of biocompatible sensors applied in the diagnosis of ischemia, rheumatoid arthritis, obesity, etc. [6,7].

HSA, present in the blood at the level of 40–50 g L^−1^, is a single non-glycosylated, *α*-chain protein consisting of 585 amino acids with a molar mass of 66,439 Da [8]. It is mainly responsible for osmotic pressure regulation and for the transport of fatty acids, ion (calcium) drugs, and hormones [8,9]. In electrolyte solutions, particularly in 0.15 M NaCl, corresponding to the blood serum environment, the molecule exhibits positive electrophoretic mobility at a pH lower than 5.0, while at a pH higher than 7.4, it exhibits negative mobility [10]. It belongs to the class of rather stable globular proteins characterized by an irregular shape approximated by a triangular model [11,12]. In ref. [11], a coarse-grained (bead) model of albumin molecule was developed, enabling an effective description of its adsorption mechanism in terms of the random sequential adsorption (RSA) approach [13].

Numerous experimental investigations aimed at elucidating the mechanism and kinetics of HSA adsorption at solid/electrolyte interfaces were carried out using various experimental methods such as the ellipsometry [14,15], the total internal reflection fluorescence (TIRF) [16,17], the surface plasmon resonance (SPR) technique [18] and the optical light mode waveguide spectroscopy (OWLS) [15,19,20,21,22,23,24,25,26]. In reference [26], the streaming potential measurements calibrated by AFM and XPS were applied to determine the adsorption kinetics of HSA on mica under various ionic strengths and pHs [27]. In a few works, the kinetics of HSA on silica and gold sensors were studied using the quartz crystal microbalance (QCM) method [26,28,29].

It should be mentioned, however, that these methods exhibit several disadvantages which limit their range of applicability. For example, the ellipsometry is not sensitive enough for the low coverage limit, the OWLS and the QCM methods require expensive and often disposable sensors and the interpretation of the primary signal is not unique. This especially concerns the latter method, where the frequency shift is controlled by the hydrodynamic flow pattern rather than by the adsorbed protein mass [30,31,32,33]. The TIRF method needs the investigated proteins to be fluorescently labeled that may change their structure. Similarly, the sensitivity of the streaming potential method decreases for the larger protein coverage range, especially if they bear a low surface charge. Because of the limitations of these techniques, the understanding of albumin adsorption at solid substrates, especially when modified by the adsorption of polyelectrolytes, still remains incomplete.

Therefore, the goal of this work is to quantitatively determine the adsorption of albumin on bare silica substrate for a broad range of pHs using the reflectometry method, which has been applied for thorough kinetic investigations of nanoparticle adsorption [34,35,36]. The methods exhibit pronounced advantages, enabling precise in situ measurements of adsorption/desorption kinetics on silica substrates, without relying on commercial supplied sensors. As to our knowledge, this method was only sporadically used for protein adsorption studies, albeit without any quantitative interpretation of the obtained data [35].

Results obtained for the bare silica are exploited as reference state for the interpretation of the HSA adsorption kinetics on the poly-L-arginine (PARG) functionalized silica, which is the main goal of this work. This polyelectrolyte has gained a special interest among researchers because of its unique properties and its environmental friendly and biocompatible behavior [37,38]. It represents a cationic biopolymer composed of polymerized physiologically active L-arginine amino acid. On the molecular level, L-arginine contains a positively charged *α*-amino group, an *α*-carboxylic acid group and a 3-carbon aliphatic straight-chain-ended-with-a-guanidine group, constituting a side chain of the monomer structure [39]. Under physiological pH, the carboxylic acid is deprotonated (–COO^−^) (pKa > 2.2) [40], the *α*-amino group is protonated (–NH_3_^+^) (pKa > 9.0) and the guanidine group is also protonated, resulting in guanidinium form (–C–(NH_2_)_2_^+^), which exhibits a pKa higher than 12.5 [39,40,41,42].

## 2. Materials and Methods

### 2.1. Materials

The experimental data presented hereafter were obtained for the human serum albumin (HSA) supplied in the form of a lyophilized powder 99% (Sigma-Aldrich, St. Louis, MO, USA) with a nominal fatty acid content of 0.02% and poly-L-arginine hydrochloride (PARG), a synthetic polyamino acid with an average molar mass of 42 kg mol^−1^ (Sigma-Aldrich).

All chemical reagents such as sodium chloride and hydrochloric acid were commercial products of Sigma-Aldrich and were used without additional purification. Ultrapure water was obtained using the Milli-Q Elix and Simplicity 185 purification system from Millipore (Merck Group, Darmstadt, Germany).

The ionic strengths of the studied solutions were adjusted using NaCl and PBS, respectively. For NaCl, the solution of desired ionic strength (10 mM or 150 mM) was prepared using 1 M NaCl stock solution and ultrapure water. In the case of the buffer solution, phosphate-buffered saline (0.138 M NaCl and 0.0027 M KCl, 0.01 M Na_2_HPO_4_, 0.0018 M KH_2_PO_4_) was purchased from Sigma-Aldrich and dissolved in 1 L of ultrapure water in order to obtain the solution exhibiting an ionic strength of 150 mM and pH 7.4.

The pHs for NaCl solutions were adjusted with 0.1 M HCl and 0.1 M NaOH, respectively.

The effective bulk concentration of albumin molecules in the stock solutions was spectrophotometrically determined according to the procedure described by Kujda et al. [43]. On the other hand, the bulk concentration of the diluted albumin solutions (1–20 mg L^−1^) used in the QCM measurements was determined by AFM imaging and the enumeration of single molecules adsorbed at mica [27].

### 2.2. Methods

Primary physicochemical properties of HSA molecules comprising the electrophoretic mobility (*μ_e_*) and the diffusion coefficient (*D*) were measured by the laser doppler velocimetry (LDV) and dynamic light scattering (DLS) methods, respectively, using the Zetasizer Nano ZS apparatus (Malvern Panalytical, Malvern, United Kingdom). The hydrodynamic diameter (*d_H_*) was calculated using the Stokes–Einstein relationship:(1)$$d_{H}=\frac{kT}{3\pi \eta D}$$
where *k* is the Boltzmann constant, *T* is the absolute temperature and *η* is the dynamic viscosity of the solution. The zeta potential (*ζ*) of the molecules was calculated using the Henry formula:(2)$$\zeta =\frac{3\eta \mu_{e}}{2\varepsilon f(\kappa d_{H})}$$
where *ε* is the electric permittivity of the electrolyte and *f*(*κd_H_*) is the Henrys correction function.

The kinetics of albumin adsorption was determined using the optical reflectometry with a stagnation point flow cell. A home-built fixed-angle reflectometer was equipped with the following components: a polarized green diode laser working with a wavelength of 532 nm (World Star Tech TECGL–532 Series, Markham, ON, Canada); a home-built flow through cell consisting of a quartz prism with an inlet borehole of *r_b_* = 0.5 mm; a spacer of *h* = 0.85 mm, ensuring a horizontal gap between the surface and the prism; and a detector with a beam splitter and two diodes acquiring perpendicular (*R_s_*) and parallel (*R_p_*) components of the reflected beam. The syringe pump was used to pump a solution through the cell with the regulated flow rate. The dry mass of the deposited species, *Γ(t)*, can then be calculated from the formula:(3)$$\Gamma \left(t\right)=\frac{1}{B}\cdot \left(\frac{S\left(t\right)-S\left(0\right)}{S\left(0\right)}\right)$$
where *B* is the sensitivity constant, and *S*(*t*) and *S*(0) are the time-dependent reflectometry signal and its initial value signal, respectively, calculated as:(4)$$S=\frac{R_{p}}{R_{s}}$$

The sensitivity constant was obtained theoretically using a homogeneous slab model, assuming that the silicon block is coated with silica layer and deposited macromolecular layers [44,45]. The refractive indices of silicon and silica, used for calculation, were 4.132 + 0.033i and 1.461, respectively [46,47], whereby the corresponding refractive indices for deposited macromolecules were calculated from the dependence:(5)$$n=n_{w}+\frac{\Gamma }{H}\cdot \frac{dn}{dc}$$
where *n_w_* = 1.335 is the refractive index of water, *H* is the thickness of the additional layer and *dn*/*dc* is the refractive index increment. The values of refractive index increment used in a model were 0.142 and 0.182 mL g^−1^ for PARG and HSA, respectively [48,49]. The obtained sensitivity constant *B* values were 0.021 and 0.026 m^2^ mg^−1^ for PARG and HSA, respectively. Both values were constant within a whole range of deposited masses. More details concerning the experimental setup and data analysis can be found elsewhere [34,50].

The human serum albumin adsorption was studied for two surfaces: (1) bare silica and (2) PARG-modified silica surface.

The PARG-covered surface was prepared in situ in the reflectometer. The oxidized silicon wafer was mounted in the cell and rinsed with a 150 mM NaCl pH 5.8 electrolyte solution. Subsequently, the PARG solution of 0.5 mg L^−1^ in the same electrolyte was flushed over 30 min to adsorb a saturated PARG layer over the silica surface. Finally, the pure electrolyte solution was flushed for 20 min. Subsequently, the HSA solution in the same electrolyte solution was injected. Once the saturated layer was obtained, the cell was rinsed again with the HSA-free electrolyte solution.

In the case of HSA adsorption on bare silica, the first stage, regarding PARG deposition, was omitted.

The zeta potential of the bare silica substrate was determined using the streaming current technique with SurPASS (Anton Paar GmbH, Graz, Austria). Two surfaces of the size of 10 mm × 20 mm, carefully cleaned before measurement, were mounted onto the instrument on both sides of the flow cell. The streaming current *I* was measured as a function of the pressure difference Δ*P*, which induced the electrolyte to flow through the cell. The electrokinetic potential *ζ* was calculated from the Helmholtz–Smoluchowski relationship [51,52]:(6)$$\zeta =\frac{\eta L}{\varepsilon S_{c}}\left(\frac{\Delta I}{\Delta P}\right)$$
where *η* = 0.89 mPas is the dynamic viscosity of water, *L* = 20 mm is the length of the rectangular channel, and *S_c_* = 1 mm^2^ is the perpendicular cross-sectional area of the channel.

The zeta potential of the PARG-covered silica substrates was determined using the streaming potential technique, which enabled in situ measurements of the polycation layer formation. The investigations were carried out using a homemade, four-electrode microfluidic cell according to the previously described procedure [27]. Primarily, the streaming potential, *E_s_*, was measured as a function of the hydrostatic pressure difference, Δ*P*. Using the slope of this dependence, the zeta potential of the substrate was calculated from the Smoluchowski equation considering the correction for the surface conductivity of the cell, *K_e_*:(7)$$\zeta_{i}=\frac{\eta K_{e}}{\varepsilon }\left(\frac{\Delta E_{s}}{\Delta P}\right)$$

In order to prevent PARG depletion, all glassware were preconditioned three times with the polycation solutions of the same concentration as that used in the experiments.

The topographical parameters of the silica surface were determined by the ex situ AFM method under ambient conditions using the NT-MDT Solver BIO (NT-MDT Spectrum Instruments, Moscow, Russia) device with the SMENA SFC050L scanning head. The number of particles per unit area (typically one square micrometer), denoted hereafter by *N*, was determined by a direct counting of over at least five equal-sized areas randomly chosen over the sensor.

All experiments were performed at a temperature of 298 K.

## 3. Results and Discussion

### 3.1. Bulk HSA and Substrate Characteristics

The dependence of the electrophoretic mobility of HSA molecules on pH determined in this way is shown in Figure 1. At pH 4.0, the zeta potentials, which were calculated from Equation (2), were equal to 1.86 and 0.95 μm cm s^−1^ V^−1^ for 10 and 150 mM NaCl concentration, respectively. At larger pHs, the electrophoretic mobility rapidly decreased and attained negative values at pH above 5.2, which can be treated as the isoelectric point of the molecules.

The diffusion coefficient of HSA molecules at pH 4.0 was equal to 6.3 ± 0.2 × 10^−7^ cm^2^ s^−1^ independently of NaCl concentration that corresponded to the hydrodynamic diameter equal to 7.8 ± 0.2 nm. However, at pH 5–6 the hydrodynamic diameter monotonically increased with time, indicating that HSA solutions were unstable.

Thorough characteristics of the silica substrate, used in the adsorption kinetic measurements, comprising its topography and the zeta potential were acquired using the atomic force microscopy (AFM) and electrokinetic methods (streaming current/potential) described above. A typical AFM scan of the silica substrate surface with the corresponding height profiles is shown in Figure 2.

Using these primary data, the three main parameters characterizing the surface topography, i.e., the average surface height, $\bar{h}$, the root mean square (*rms*) and the skewness were calculated as follows [53]:(8)$$\bar{h}=\frac{1}{N_{i}}\sum_{i=1}^{N_{i}}\left(h_{i}-h_{0}\right)\;rms^{2}=\frac{1}{N_{i}}\sum_{i=1}^{N_{i}}{\left(h_{i}-\bar{h}\right)}^{2}\;sk=\frac{1}{rms^{3}N_{i}}\sum_{i=1}^{N_{i}}{\left(h_{i}-\bar{h}\right)}^{3}$$
where *h_i_* denotes the surface height at the consecutive scan nodes, *h*_0_ is the reference (minimum) height and *N_i_* is the number of pixels corresponding to the AFM scanning area.

The *rms* and the skewness of the silica substrate determined in this way were 0.15 ± 0.01 nm and 0.32 ± 0.02, which correspond to the maximum surface height of 0.4 nm calculated assuming that the surface roughness can be approximated by a spheroidal cap shape [53]. A comparison of these data with the typical silica QCM sensor *rms* of 1 nm indicates that the substrate used in our investigations was significantly smoother [28,29].

The zeta potential of the bare silica substrate was determined by streaming current measurements according to the procedure described above. At pH 4.0, it was equal to −18 and −20 mV for NaCl concentrations of 150 and 10 mM, respectively. At pH 7.4, it decreased to −35 and −65 mV for the same NaCl concentrations (see Figure 3A).

The zeta potential of the PARG-covered silica substrates was derived from the streaming potential measurements, which enabled in situ measurements of the polycation layer formation carried out at pH 5.6 and the volumetric flow rate of 0.35 cm^3^ s^−1^. The bulk concentration of PARG in 10 mM NaCl was equal to 5 mg L^−1^ and the adsorption time was 30 min. Under such conditions, a stable monolayer of PARG was formed. The results shown in Figure 3B confirm that the zeta potential was positive and assumed average values of 35 and 24 mV for 10 and 150 mM NaCl concentration for the pH range 3.5 to 9, respectively.

The zeta potential vs. pH dependencies shown in Figure 3A,B for bare and PARG-covered silica enabled a proper interpretation of its adsorption kinetic on the substrates presented in the next section.

### 3.2. Kinetics of HSA Adsorption on Bare Silica Sensor

The first series of measurements was devoted to determining the kinetics of HSA adsorption on the bare silica, primarily to calibrate the optical reflectometry method in order to confirm its ability to furnish quantitative coverage data. Such calibration kinetic runs performed at pH 4.0 and 10 mM NaCl are shown in Figure 4 as the dependence of the albumin coverage on the adsorption time. The adsorption was carried out over the prescribed time period (20 to 1200 s). Afterward, the desorption run was initiated where the solvent (electrolyte solution) was flushed through the cell. Finally, the silica substrate was carefully washed and examined by AFM under ambient air conditions. The number of adsorbed albumin molecules over equally sized surface areas was determined according to the previously applied procedure [13,29] and expressed as the surface concentration, i.e., the number of molecules per a square micrometer, denoted by *N*. In order to facilitate the comparison with the results derived from reflectometry, the surface concentration of molecules was recalculated for the mass coverage, expressed in mg m^−2^, using the following formula:(9)$$\Gamma =10^{15}\frac{M_{w}}{N_{Av}}N$$
where *M_w_* is the molar mass of HSA expressed in kg mol^−1^, *N_Av_* is the Avogadro number and *N* is expressed in molecules per μm^−2^.

The results obtained in the experiments shown in Figure 4 indicate that the albumin coverage determined by reflectometry agreed within experimental error bounds with the AFM coverage derived from the tedious counting of the adsorbed molecules. This validates the use of reflectometry as an effective tool for a convenient determination of protein adsorption kinetics.

In the next step, a series of experiments were carried out with the aim to determine the mass transfer rate constants and the maximum coverage of albumin on the bare silica substrate for different NaCl concentrations and pHs. In Figure 5, the results of the kinetic experiments performed at pH 4.0 and 10 mM NaCl (for the flow rate of 1.66 × 10^−2^ cm^3^ s^−1^) are shown, where the influence of the bulk concentration of albumin was investigated. As can be seen, even for the lowest HSA concentration of 0.1 mg L^−1^ that corresponds to the nM concentration range, a well-pronounced kinetic run was recorded. This confirms a large sensitivity of the reflectometric method. For increasing bulk albumin concentration, the initial deposition (defined as the derivative of the coverage upon the time) systematically increased, and, in consequence, for *c_b_* larger than 1 mg L^−1^, the entire adsorption run was completed within 60 s. In order to quantitatively estimate the precision of the measurements, the initial adsorption rate was plotted against the bulk HSA concentration (see Figure 5).

This dependence was well described by a linear function with the slope of 5.3 ± 0.2 × 10^−3^ L m^−2^s^−1^ = 5.3 ± 0.2 × 10^−4^ cm s^−1^, which corresponds to the mass transfer rate constant *k_c_*. It should be underlined that the results shown in Figure 5 can be used to precisely determine the albumin bulk concentration within the nM range and for a measurement time of only 1 min. This represents a considerable advantage over the spectrophotometric methods, which only yields a reasonable precision for albumin bulk concentration above µM.

In Figure 6A, a series of kinetic runs obtained at pH 4.0, 150 mM NaCl for various flow rates are presented. One can observe that the plateau coverages of albumin attained after an adsorption time of 900 s were fairly independent of the flow rate with an average value of 1.1 mg m^−2^. The consecutive desorption runs, where the electrolyte solution was flushed through the cell, confirmed that the plateau coverages did not change, thereby suggesting a negligible desorption of albumin independently of the flow rate.

Except for the maximum coverage, the mass transfer rate constants for various solution flow rates were also determined using the short-time kinetic runs—presented for the sake of convenience in Figure 6B as a dependence of the derivative of the adsorption rate upon time. They are collected in Table 1 and compared with the theoretical values calculated from the formula derived in ref. [54], assuming an irreversible adsorption mechanism of the molecules [54]:(10)$$k_{c}=C\left(r_{b}/r_{t}\right)\frac{\alpha^{1/3}Q^{1/3}D^{2/3}}{r_{t}^{4/3}}$$
where *α_r_* is the dimensionless parameter depending on the Reynolds number *Re*. The *C*(*r_b_*/*r_t_*) function is defined as
(11)$$C\left(r_{b}/r_{t}\right)=0.320\int_{0}^{r_{b}/r_{t}}\frac{{\left(\sin\bar{x}\right)}^{\pi /2}}{{\left[\int_{0}^{\bar{x}}{\left(\sin y\right)}^{\left(3\pi /2\right)-1}dy\right]}^{1/3}}$$
where $\bar{x}$ = *πr*/2*r_i_* is the normalized tangential coordinate, *r* is the radial distance from the cell center, *r_t_* is the radius of the inlet tubing and *r_b_* is the radius of the laser beam.

As can be seen, the theoretical and experimental data agree with each other for all the flow rates studied. This fact indicates that the initial absorption rate of albumin at pH 4.0 attained maximum values pertinent to single molecules under the irreversible mechanism.

This conclusion is further supported by the data shown in Table 2, where the maximum coverages determined in our work by reflectometry, equal to 0.6 ± 0.03 and 1.1 ± 0.05 mg m^−2^, for 10 and 150 mM NaCl, respectively, well correlated with the theoretical coverages, calculated by applying the coarse-grained random sequential adsorption (RSA) modeling [13] and with the previous experimental results obtained by OWLS [28] and the streaming potential methods [27].

Analogously as in previous works [27,29], the increase in the maximum coverage of HSA on the bare silica substrate with the electrolyte concentration can be interpreted in terms of the decreasing range of the lateral electrostatic interactions among adsorbed molecules.

Adsorption kinetics of albumin on the bare silica substrate at larger pHs were also investigated. The results shown in Figure 7 indicate that it was considerably less efficient compared to pH 4.0 both in respect to the initial rate and the maximum coverage. Thus, at pH 4.8, the albumin coverage of 0.8 mg m^−2^ was attained after a time of 800 s, whereas at pH 5.4, the coverage was 0.2 mg m^−2^ for the same adsorption time. At pH 7.4, the coverage after 800 s was only 0.05 mg m^−2^. These results indicate that the decrease in the adsorption rate of albumin was strictly correlated with the decrease in its zeta potential (see Figure 1), which confirms the electrostatic interaction-driven mechanism. These results also indicate that in order to achieve a significant adsorption of albumin under a physiological pH of 7.4, the silica substrate zeta potential should be converted to positive, which can be realized by a controlled PARG adsorption (see Figure 3). The results obtained in this case are discussed in the next section.

### 3.3. Kinetics of HSA Adsorption on PARG-Modified Silica Surface

In Figure 8, the influence of pH on the adsorption kinetics of HSA on the PARG-functionalized silica is illustrated for a NaCl concentration of 150 mM. These results indicate that the adsorption at pH of 7.4 (regulated by the addition of NaOH) was the most efficient, characterized by a rapid initial increase and the plateau value of 1.5 mg m^−2^, which is 0.4 mg m^−2^ larger than the maximum coverage of HSA on the bare silica. At this pH, the zeta potentials of albumin and the substrate were of the opposite sign, equal to −20 and 20 mV, respectively. At lower pHs, the initial deposition rate and the plateau value of the coverage systematically decreased; the latter was only 0.1 mg m^−2^ at pH 4.0. At this pH, the zeta potentials of albumin and the substrate were of the same sign and equal to 20 mV. Therefore, these results, analogously as before, confirm the electrostatic interaction-controlled mechanism of HSA adsorption.

Unexpectedly, at pH 7.4, stabilized by the PBS buffer, the HSA adsorption kinetics were significantly less efficient than for the above case of pH 7.4 but stabilized by NaOH. This result suggests that a specific interaction of the buffer components (negatively charged phosphate-buffered saline) with the positively charged PARG molecules appeared in this case. This effect can explain a significant spread in the maximum coverages of albumin reported in the literature.

In order to better illustrate the trends pertinent to HSA adsorption unveiled in this work, the results obtained for the bare and PARG-functionalized silica are shown in Figure 9, as the dependence of the plateau value of the albumin coverage normalized to the maximum coverage obtained at pH 4.0 for bare silica (equal to 1.1 mg m^−2^). As can be seen in this figure, an effective albumin adsorption at pH 7.4 was only feasible at the PARG-functionalized silica.

## 4. Conclusions

Thorough HSA adsorption kinetic experiments performed using reflectometry calibrated by single-molecule AFM imaging showed that this method yields quantitative data for the nanomolar protein concentration in the solution. This represents a considerable advantage over the spectrophotometric methods, which only yield a reasonable precision for albumin bulk concentration above a micromolar range.

Our measurements also enabled us to determine the mass transfer rate constants pertinent to the albumin adsorption under various flow rates and electrolyte concentrations. These experimental results agreed with the theoretical date derived by applying the hybrid random sequential adsorption approach. This fact was interpreted as the evidence of an irreversible adsorption mechanism of albumin and the stability of its solutions. Therefore, it was argued that these results can serve as useful reference data, enabling a robust estimation of the stability of protein solutions.

Investigations performed for various pHs, for the bare and poly-L-arginine-modified silica, showed that the adsorption effectiveness is strictly correlated with the zeta potential of HSA. This was interpreted as the evidence of a significant role of electrostatic interactions of two types: (i) the direct interactions of molecules with the substrate surface and (ii) the lateral electrostatic interactions among the molecules within the adsorbed layer that control the protein stability and the maximum coverage. In consequence of the former interactions, an effective albumin adsorption at pH 7.4, where its zeta potential became negative, was only feasible at the PARG-functionalized silica bearing a positive zeta potential at this pH.

## Figures and Tables

**Figure 1 biomolecules-14-00531-f001:**
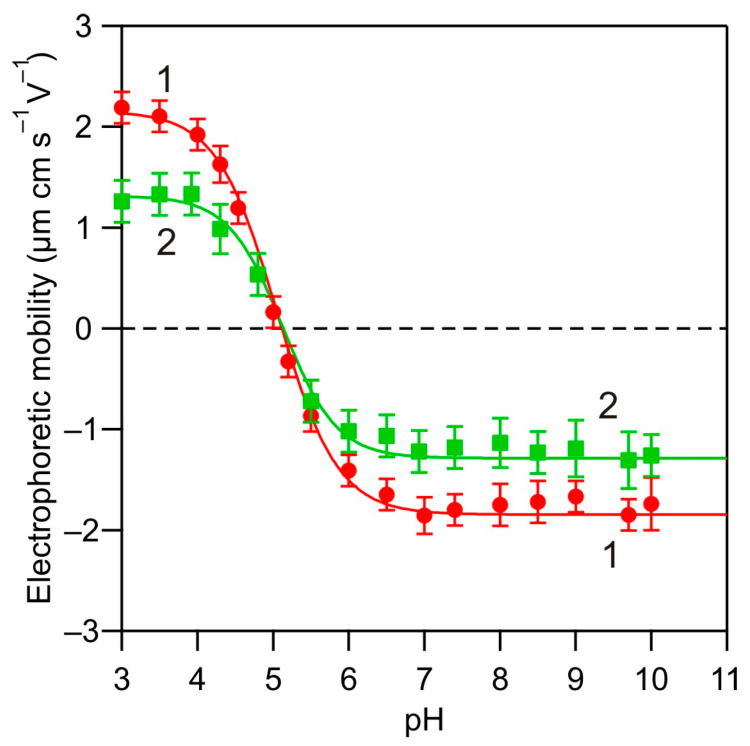
Electrophoretic mobility of HSA molecules in the bulk solutions vs. pH determined by the LDV measurements: (1) 10 mM NaCl and (2) 150 mM NaCl. The lines are a guide for the eyes.

**Figure 2 biomolecules-14-00531-f002:**
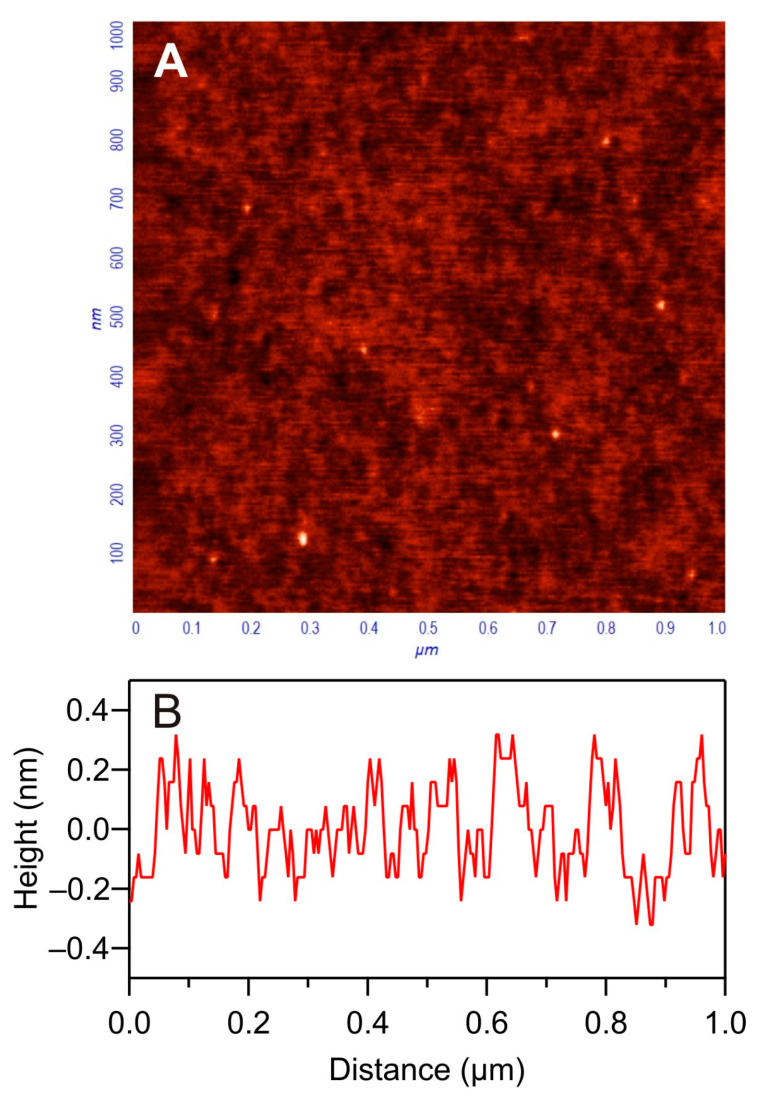
(**A**) AFM image of the silica sensor with (**B**) the height profile.

**Figure 3 biomolecules-14-00531-f003:**
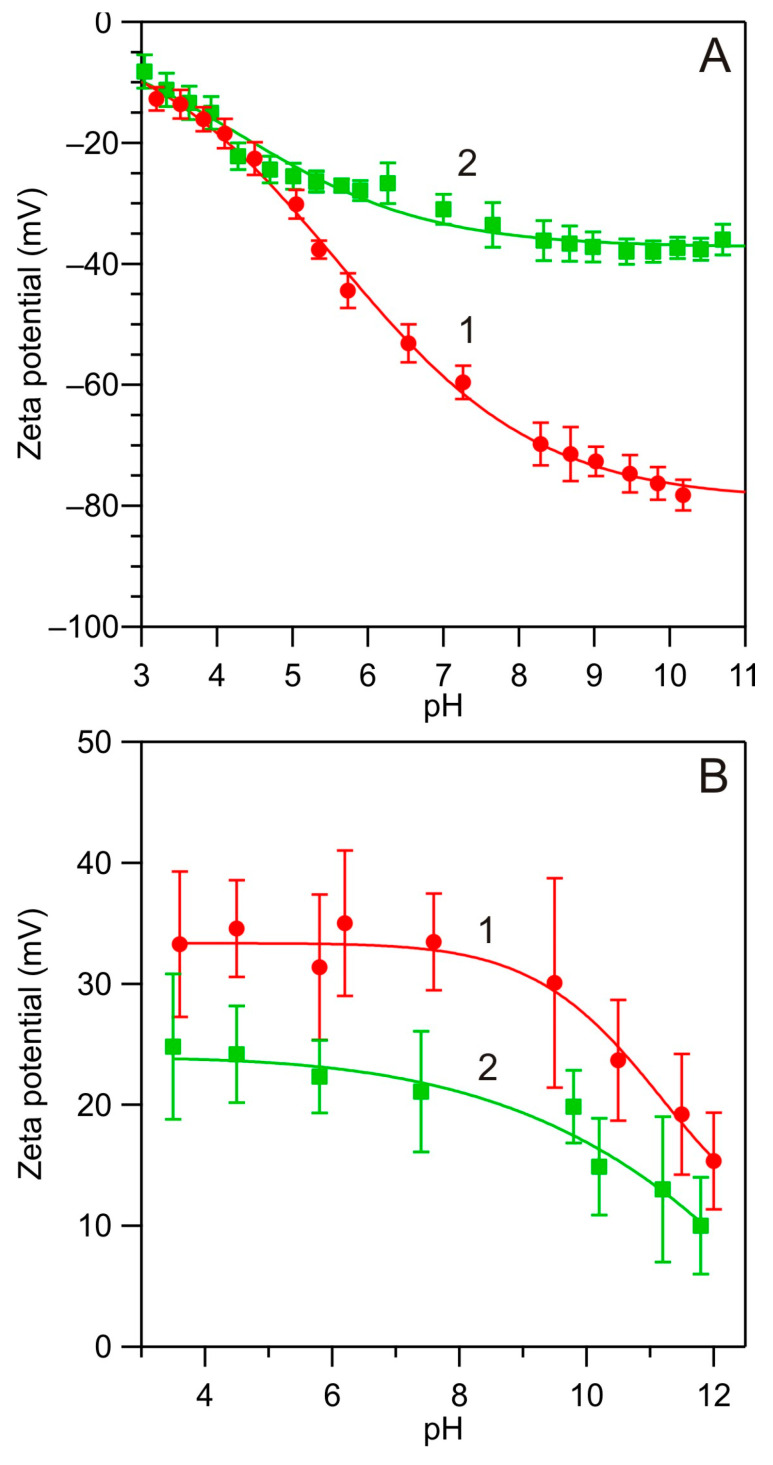
(**A**) Zeta potential of the bare silica substrate vs. pH determined by streaming current measurements: (1) 10 mM NaCl and (2) 150 mM NaCl. (**B**) Zeta potential of the PARG-covered silica substrate vs. pH determined by streaming potential measurements (microfluidic cell): (1) 10 mM NaCl and (2) 150 mM NaCl. The lines are a guide for the eyes.

**Figure 4 biomolecules-14-00531-f004:**
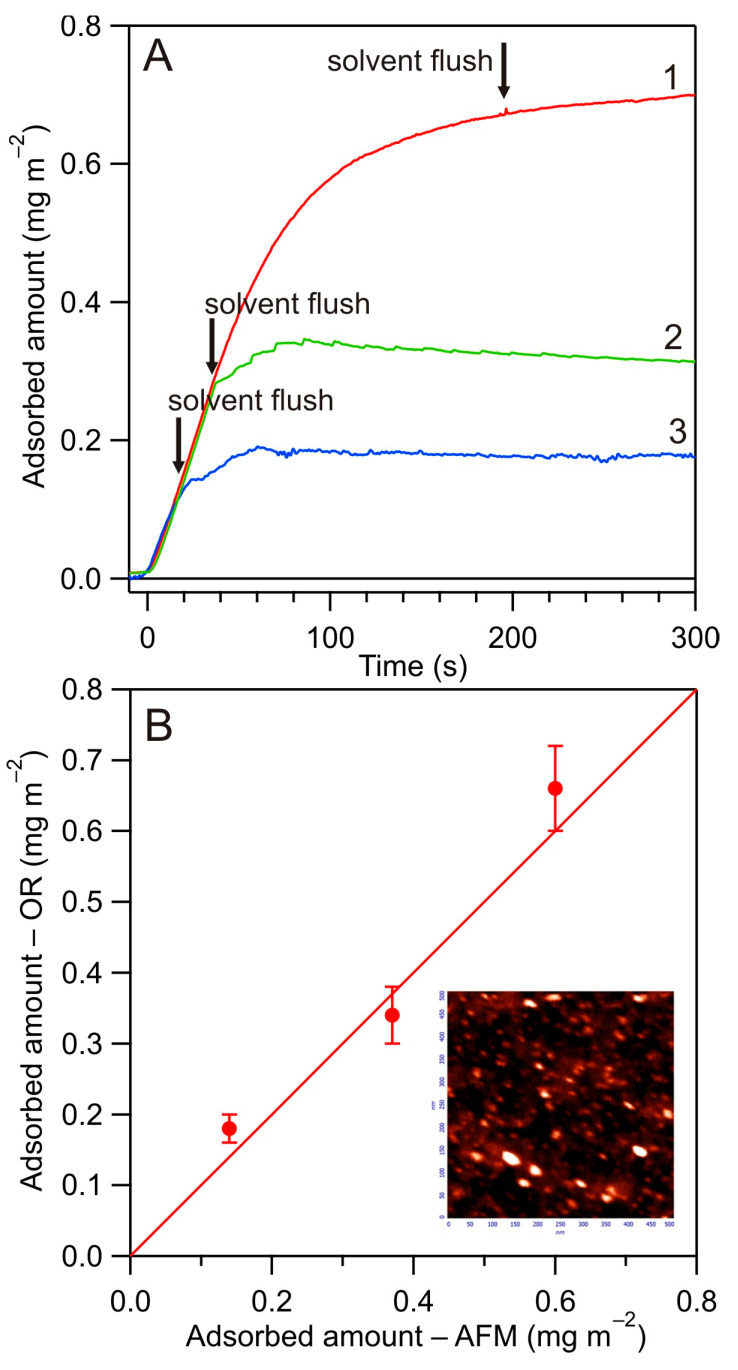
(**A**) Kinetics of HSA adsorption at the silica sensor determined by reflectometry—calibration runs performed at pH 4.0, 10 mM NaCl, bulk concentration *c_b_* = 5 mg L^−1^, flow rate *Q* = 8.3 × 10^−3^ cm^3^ s^−1^: (1) adsorption time 180 s, (2) adsorption time 40 s, (3) adsorption time 20 s. (**B**) The dependence of the HSA coverage derived from optical reflectometry (OR) vs. the coverage derived from AFM, calculated from Equation (9); the inset shows the HSA molecule layer imaged by AFM.

**Figure 5 biomolecules-14-00531-f005:**
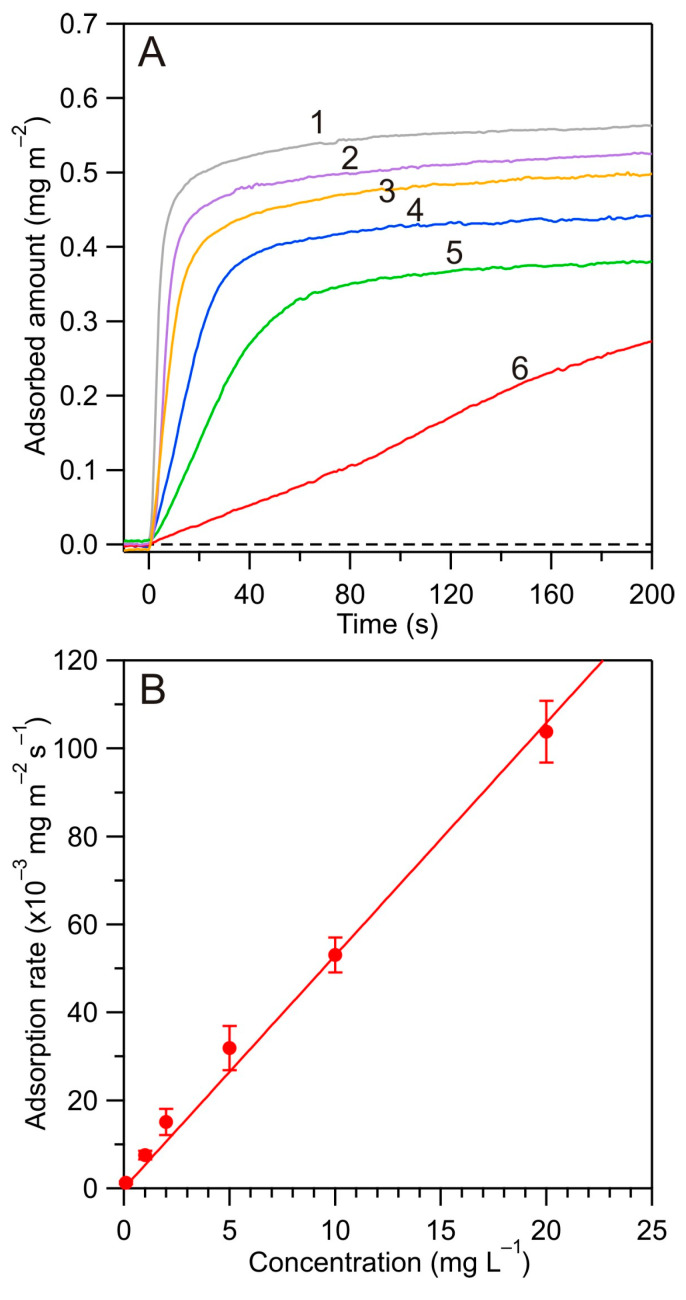
(**A**) Kinetics of HSA adsorption at the silica sensor determined by reflectometry for various bulk concentrations, pH 4.0, 10 mM NaCl, solution flow rate *Q* =1.66 × 10^−2^ cm^3^ s^−1^: (1) *c_b_* = 20 mg L^−1^, (2) *c_b_* = 10 mg L^−1^, (3) *c_b_* = 5 mg L^−1^, (4) *c_b_* = 2 mg L^−1^, (5) *c_b_* = 1 mg L^−1^, (6) *c_b_* = 0.1 mg L^−1^. (**B**) The initial adsorption rate vs. the HSA bulk concentration for the same experimental conditions as in (**A**). The line shows the linear fit of the experimental data.

**Figure 6 biomolecules-14-00531-f006:**
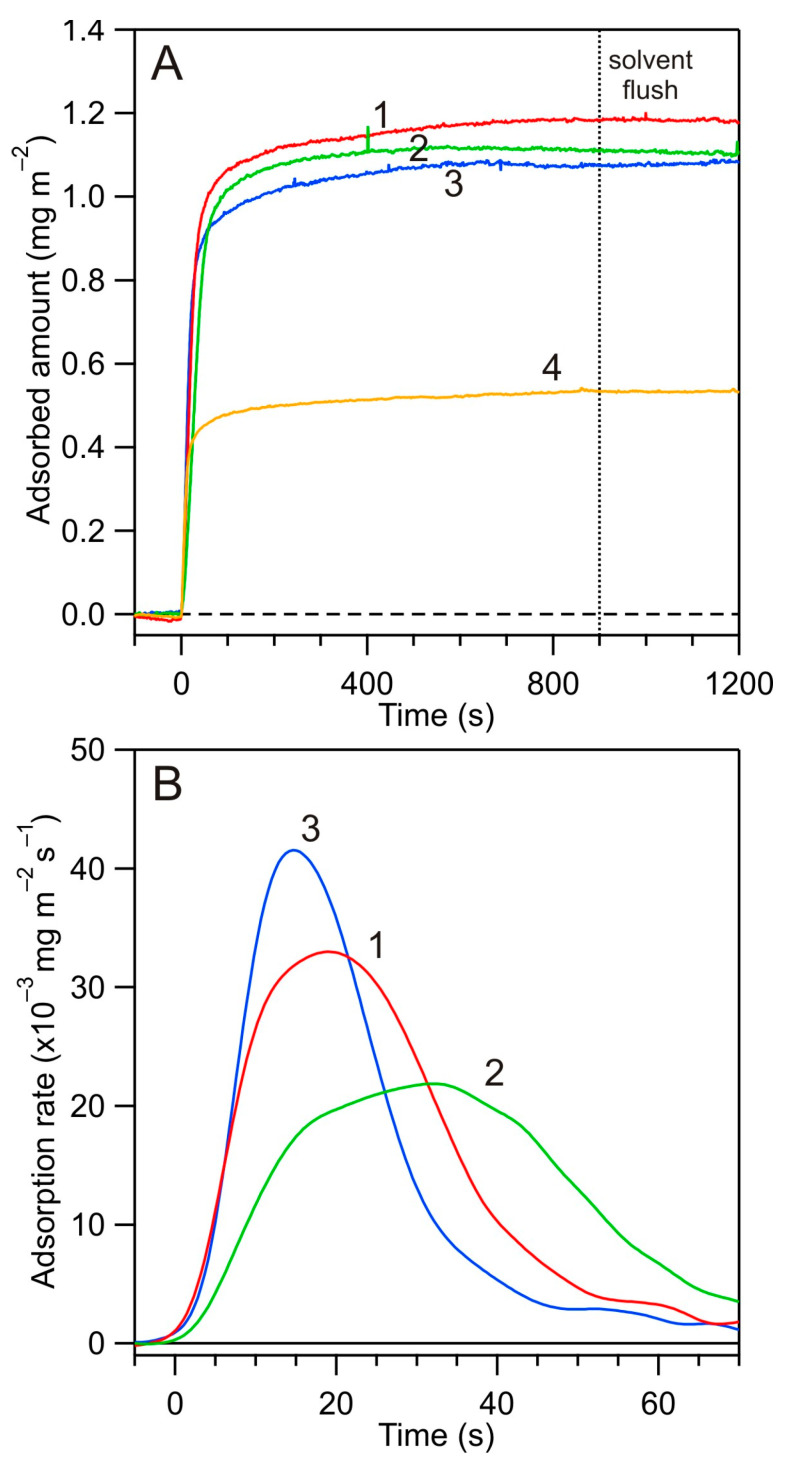
(**A**) Kinetics of HSA adsorption at the silica sensor determined by reflectometry for various flow rates, pH 4.0, *c_b_* = 5 mg L^−1^: (1) *Q* = 8.33 × 10^−3^ cm^3^ s^−1^, 150 mM NaCl, (2) *Q* = 3.33 × 10^−3^ cm^3^ s^−1^, 150 mM NaCl, (3) *Q* = 1.66 × 10^−2^ cm^3^ s^−1^, 150 mM NaCl, (4) *Q* = 1.66 × 10^−2^ cm^3^ s^−1^, 10 mM NaCl. (**B**) The derivative of the adsorption amount upon time for various flow rates for the same experimental conditions as in (**A**): (1) *Q* = 8.33 × 10^−3^ cm^3^ s^−1^, 150 mM NaCl, (2) *Q* = 3.33 × 10^−3^ cm^3^ s^−1^, 150 mM NaCl, (3) *Q* = 1.66 × 10^−2^ cm^3^ s^−1^, 150 mM NaCl.

**Figure 7 biomolecules-14-00531-f007:**
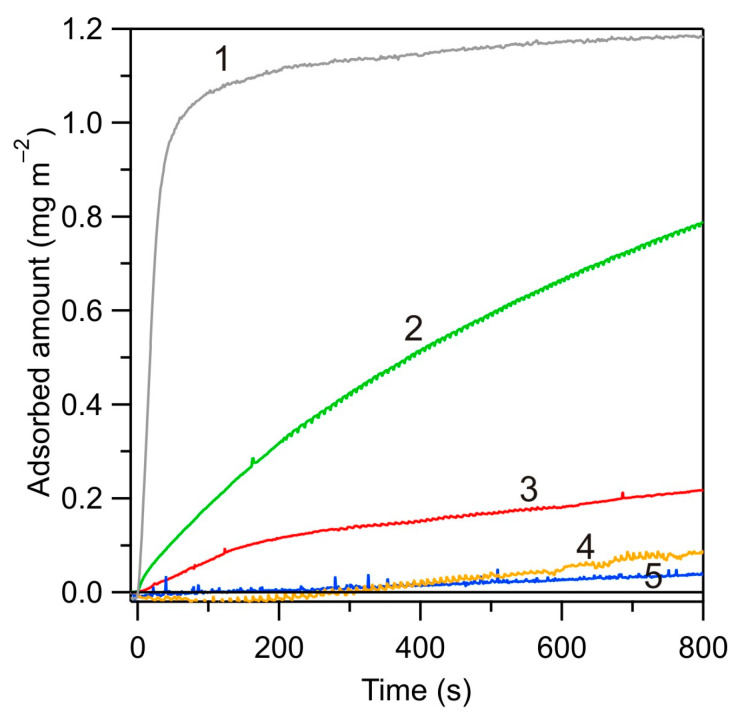
Kinetics of HSA adsorption on bare silica sensor determined by reflectometry for various pHs, albumin bulk concentration *c_b_* = 5 mg L^−1^, *Q* = 8.33 × 10^−3^ cm^3^ s^−1^, 150 mM NaCl: (1) pH 4.0, (2) pH 4.8, (3) pH 5.4, (4) pH 7.4, and (5) pH 8.6.

**Figure 8 biomolecules-14-00531-f008:**
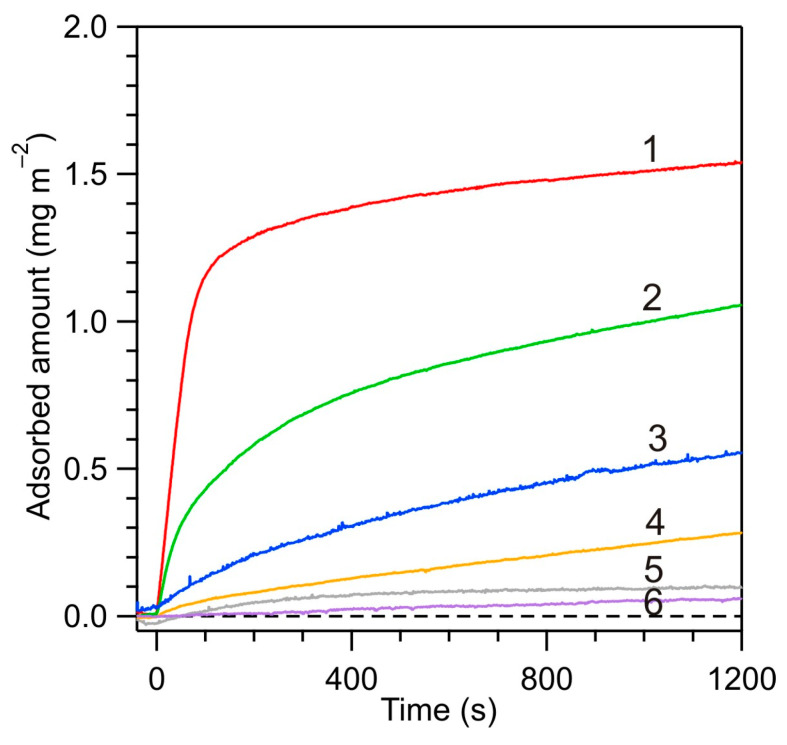
Kinetics of HSA adsorption on PARG-covered silica sensor determined by reflectometry for various pHs, albumin bulk concentration *c_b_* = 5 mg L^−1^, 150 mM NaCl, *Q* = 8.33 × 10^−3^ cm^3^ s^−1^: (1) pH 7.4 (NaCl), (2) pH 8.6, (3) pH 7.4 (PBS); (4) pH 5.8; (5) pH 4.0, and (6) pH 3.5.

**Figure 9 biomolecules-14-00531-f009:**
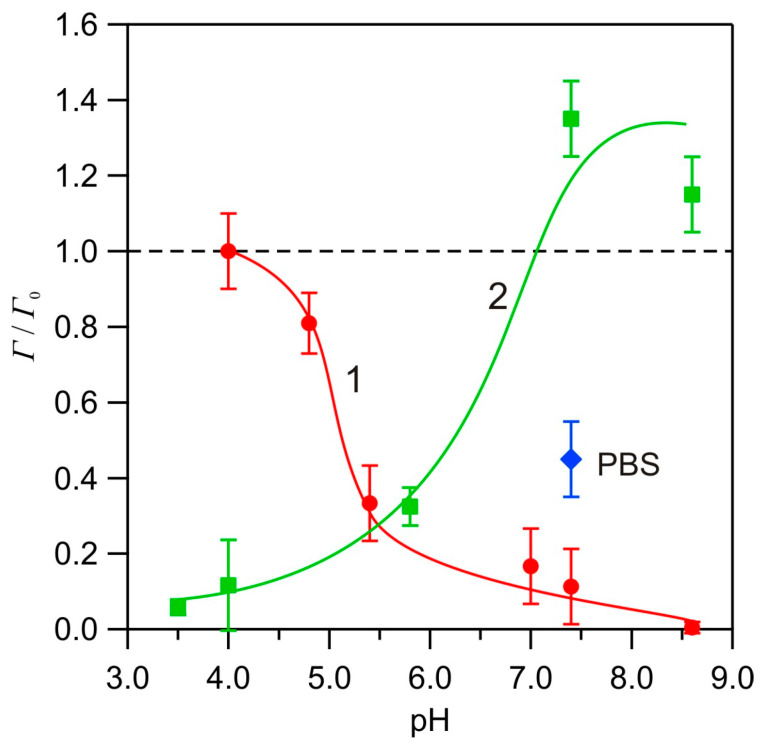
Dependence of the plateau coverages of HSA (normalized to the maximum coverage of 1.1 mg m^−2^ for the bare silica) for 150 mM NaCl on pH: (1) adsorption on bare silica sensor and (2) adsorption on PARG-covered sensor. The blue point presents experimental data for PBS solution. The lines are guides for the eyes.

**Table 1 biomolecules-14-00531-t001:** The mass transfer rates for HSA molecules on the bare silica sensors determined by reflectometry for various flow rates, pH 4.0, 150 mM NaCl.

Flow Rate [cm^3^ s^−1^]	Reynolds Number	The *α_r_* Parameter	Mass Transfer Rate Constant Theory [cm s^−1^]	Mass Transfer Rate Constant Exp. [cm s^−1^]
1.66 × 10^−2^	12	6.8	8.2 × 10^−4^	8.1 ± 0.4 × 10^−4^
8.33 × 10^−3^	5.9	3.4	5.7 × 10^−4^	6.4 ± 0.3 × 10^−4^
3.33 × 10^−3^	2.4	2.5	3.8 × 10^−4^	4.0 ± 0.2 × 10^−4^

**Table 2 biomolecules-14-00531-t002:** The maximum coverage of HSA on the bare silica sensors determined by various methods, pH 3.5–4.0.

NaCl Concentration [mM]	* Maximum Coverage [mg m^−2^]	** Maximum Coverage [mg m^−2^]	Maximum Coverage This Work [mg m^−2^]	*** Maximum Coverage Theoretical [mg m^−2^]
1.0	0.50 ± 0.1	0.42	–	0.35
10	0.65 ± 0.1	0.66	0.60 ± 0.05	0.62
150	1.30 ± 0.1	1.30	1.10 ± 0.05	1.20

* Ref. [28]: silica sensor—the OWLS method. ** Ref. [27]: streaming potential method—adsorption on mica. *** Theoretical results calculated for single-molecule adsorption using the effective hard particle—random sequential adsorption (RSA) model [13].

## Data Availability

The data that support the findings of this study are available from the corresponding authors upon reasonable request.

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
