# Peer review of "Kinetics of Human Serum Albumin Adsorption on Polycation Functionalized Silica"

_biomolecules, 2024, doi:10.3390/biom14050531_

Round 1

Reviewer 1 Report

Comments and Suggestions for Authors

This paper by Nattich-Rak et al. reports a very thorough investigation into the adsorption of human serum albumin at native and PARG-modified SiO2 surfaces. The authors provide a detailed characterization of protein and surfaces, after which they employ reflectometry calibrated by AFM to quantitatively study the protein adsorption kinetics at these surfaces in situ and at different ionic strengths and pH values. This is a nice paper that not only provides important insights into HSA adsorption but also demonstrates the advantages of the reflectometry technique. I suggest publication in biomolecules after the following minor issues have been addressed.

1. Page 3, first paragraph: I find it a bit confusing that the authors apparantly used PBS as the electrolyte but refer to it throughout the paper as 150 mM NaCl. Also, it is not mentioned how the 10 mM NaCl solution was prepared (presumably by dilution of the stock PBS?). Or did they use both, pure NaCl as well as PBS? There is also no description how the pH was adjusted. This is particularly important with respect to comment 8 below. Please revise thoroughly.

2. Figure 2B: Shouldn't the height axis be in nm? A pm height variation seems quite challenging even for a polished wafer.

3. Figure 3A: I believe the green data should be labelled as 2 and the red data as 1, as in the other plots. Please correct.

4. Lines 218-219: The zeta potential data are shown only in Figures 1 and 3; Figure 2 shows AFM data. Please correct.

5. Lines 236-237: I believe N should be expressed in molecules µm-2 and not in mg m-2. Please correct.

6. Figure 4A: If the only difference between the curves is the adsorption time, then why do curves 2 and 3 start to saturate already before the solvent flush? Shouldn't they both coincide with curve 1 and saturate only after flushing? Please add an explanation.

7. Figure 6B: According to the y axis, the adsorption rate is shown, while the captions says it's the derivative of the adsorption rate. I believe it's rather the derivative of the adsorbed amount. Please correct.

8. Lines 359-364: I don't understand this comparison. How was the pH in both electrolytes adjusted and what is the ionic strength in both cases? Adjusting the pH with NaOH provides additional ions, could that be a contributing factor? Please provide more details of the individual samples and measurements, along with a more detailed discussion.

9. Figure 9: Please indicate which surface the value for PBS corresponds to.

Comments on the Quality of English Language

English is mostly fine. Some sentences need some minor rephrasing.

Author Response

Please find our reply in the attached document.

Reviewer 2 Report

Comments and Suggestions for Authors

This work presents the use of reflectometry as a technique for the determination of albumin adsorption kinetics on silica and silica surfaces modified with a polyelectrolyte. The work presents this technique as an interesting, fast and sensitive alternative in adsorption studies and for the development of possible sensors with a sensitivity in the nanomolar range. Despite the interest of the work, I consider that certain aspects should be reviewed in depth before publication:

Major issues

-In figure 4 A, it is not clear how it is possible that before solvent flush, the “adsorbed amount” (Y) tends to stabilise and deviates from the initial behaviour when the flow conditions and HSA concentration are the same. Only after solvent flush would one expect differences, never before. It is necessary to revise the results or explain the conditions that would make such differences possible.

-Figure 7 shows the decrease in the rate of adsorption of HSA on bare silica as a function of pH. As the authors indicate, both the amount adsorbed and the adsorption rate decrease considerably with respect to that observed at pH 4, due to the decrease in the pZ of the protein. This is true, as well as the fact that at pH 7.4 to achieve a considerable adsorption the substrate must be positively charged. However, taking into account the enormous decrease in adsorption kinetics (at pH 4 the maximum coverage is practically reached in 100 s, while at pH 4.8 it takes more than 800 s) a further explanation/comparison is needed.

-In figure 8, no explanation is given for the large decrease observed in the coverage at pH 8.6 of PARG silica. Furthemore, figure 9 does not include pH 8.6 data for PARG covered silica.

-The adsorption/desorption experiments could be detailed in materials and methods section. Although they are explained in the results and discussion section, including used models,etc. I consider this detracts from the discussion, which is also sparse.

Other comments:

-it is necessary to delete lines 85-86 of the introduction section.

-In materials and methods section (or in supplementary material section) the conditions for coating silica with poly-L-arginine should be explained in detail.

-Figure 5 B shows the data for a cb of 10 mg/L when in figure 5 A it did not appear. Is there any reason not to include it in the first figure 5A?

Author Response

(The authors gave the same response as above.)

Reviewer 3 Report

Comments and Suggestions for Authors

1.Introduction

5 th line of the 5 th paragraph (page 2). A space between “adsorption/” and “desorption” should be deleted to be “adsorption/desorption”.

3.Results and Discussion

3.1 Bulk HAS and substrate characteristics

2 nd line of the 2 nd paragraph (page 4). In Fig.1, the electrophoretic mobility of the HAS molecules looks larger than 1 µm cm s-1 V-1 for 150 mM NaCl concentration at pH 4.0, while it is written as “0.95 µm cm s-1 V-1”.  

The digit “0.95” may be the zeta potential calculated by formula (5)?

Figure 2 (page 5). The authors discussed the surface roughness from the AFM image. But the scanning area by AFM measurement is limited to 10 µm square. The authors should discuss the surface roughness using the data obtained from several areas.

“as shown in Fig.3 (A)” or else should be added at the end of the 7 th paragraph. (page 6)

Last line of page 6. “Fig.3” should be “Fig.3(B)”.

Author Response

Please find our response in the attached file.

Round 2

Reviewer 2 Report

Comments and Suggestions for Authors

I find the response to my comments adequate to address the concerns raised. I also value the modifications/corrections include in the manuscript.